# Vulnerability to Psychosis, Ideas of Reference and Evaluation with an Implicit Test

**DOI:** 10.3390/jcm8111956

**Published:** 2019-11-13

**Authors:** Pedro Bendala-Rodríguez, Cristina Senín-Calderón, Leonardo Peluso-Crespi, Juan F. Rodríguez-Testal

**Affiliations:** 1Personality, Evaluation and Psychological Treatment Department, University of Seville, 41018 Seville, Spain; pbendala@us.es; 2Faculty of Psychology, University of the Republic, Montevideo 11200, Uruguay; leonardo@psico.edu.uy; 3Department of Psychology, University of Cadiz, 11510 Cadiz, Spain; cristina.senin@uca.es

**Keywords:** ideas of reference, aberrant salience, psychosis, psychotic-like experiences, emotional counting Stroop

## Abstract

Background: Ideas of reference (IRs) are observed in the general population on the continuum of the psychotic phenotype (as a type of psychotic-like experiences, PLE). The instruments usually used to evaluate IRs show some problems: They depend on the cooperation of the participant, comprehension of items, social desirability, etc. Aims: The Testal emotional counting Stroop (TECS) was developed for the purpose of improving evaluation of individuals vulnerable to psychosis and its relationship with ideas of reference. The TECS (two versions) was applied as an implicit evaluation instrument for IRs and related processes for early identification of persons vulnerable to psychosis and to test the possible influence of emotional symptomatology. Method: A total of 160 participants (67.5% women) from the general population were selected (Mean (M) = 24.12 years, standard deviation (SD) = 5.28), 48 vulnerable and 112 non-vulnerable. Results: Vulnerability to psychosis was related to greater latency in response to referential stimuli. Version 4 of the TECS showed a slight advantage in identifying more latency in response to referential stimuli among participants with vulnerability to psychosis (Cohen’s *d* = 1.08). Emotional symptomatology (especially stress), and IQ (premorbid) mediated the relationship between vulnerability and IR response latency. Conclusions: The application of the implicit Testal emotional counting Stroop test (TECS) is useful for evaluating processes related to vulnerability to psychosis, as demonstrated by the increased latency of response to referential stimuli.

## 1. Introduction

Study of positive symptoms on a continuum from adapted functioning to psychotic disorders enables research to be directed at their risk or vulnerability factors [1,2]. Van Os and Linscott [3] suggested an extended psychosis phenotype (expression of vulnerability), which shares demographic, environmental, family and psychopathological characteristics with psychotic disorders. Vulnerability factors analyzed range from psychotic-like experiences (PLE), frequent in the general population (phenomenological continuity), to at-risk mental states, characterized by basic or attenuated psychotic symptoms, with higher rates of transition to psychosis (temporary continuity) [4,5,6,7,8,9].

Ideas of reference (IRs) are a type of PLE. They are self-attributions about what happens in one’s surroundings, mainly social [10], in which unimportant stimuli (gestures, looks, comments), are interpreted as directed at oneself [11]. IRs are a frequent, upsetting, but transitory PLE in the general population [12,13,14], possibly related to the control and regulation function of human social activity [15]. When IRs are frequent, but unstable (not persistent), they are considered among the basic symptoms [16], but when they become stable (persistent), they are attenuated psychotic symptoms [8]. When experienced with a negative or threatening meaning, both IR criteria are identified with the prodromal period of psychotic disorders. Therefore, an increase in the conviction and distress of IRs may be a relevant vulnerability factor for identifying psychotic disorders and prediction of severe psychopathology in the general population [17,18].

Another indicator of vulnerability related to IRs is what is called aberrant salience. This is excessive motivational meaning given to irrelevant stimuli [19,20]. A state of dopamine dysregulation causes certain environmental events to be attributed excessive meaning or motivational value [21]. Thus, IRs may be clinically significant (especially if they are stable) when social stimuli become aberrantly salient [22], mainly in vulnerable individuals [23,24].

Evaluation of IRs and aberrant salience is not exempt of difficulties. Self-informed evaluation, which is often referred to oneself, depends on the collaboration of the subject being evaluated, possible influence of emotional processes in appearance and maintenance of PLE and positive symptoms [25], comprehension of the items on the instruments, the level of self-awareness of the problem, or that the response is the socially desirable one [26].

In view of these drawbacks, discriminative tasks, preferentially implicit or non-declarative, are necessary. This could facilitate evaluation and its precision, and would be minimally invasive for the subject evaluated. In this line, the emotional Stroop (eStroop) paradigm could be relevant in identifying cognitive and emotional processes involved in psychosis [27], such as IRs and related processes.

The eStroop focuses on attentional interference [28]: Stimuli with negative or threatening mental content may require more attentional resources and more emotional activation if they coincide with the emotional state of the participant [29]. This interference can be observed in errors, omissions, or usually an increase in latency in the response to the stimulus presented. Attentional interference has been widely explored in psychosis [30]. An increase in response latency has been corroborated, for example, with facial stimuli interpreted as a threat to the self [31] by both delusional and vulnerable patients [32], although other studies suggest that this processing is more conscious and not automatic when facial stimuli, which are more complex than usual in the Stroop paradigm, are used [33].

In psychosis, the eStroop has usually been used to analyze persecutory content, suggesting that more interference of threatening words is associated with positive symptoms [34]. However, in a study on words with referential content versus neutral words based on the eStroop paradigm, no significant differences were found in response latency between patients (mood, personality, and psychotic disorders) and university students [35]. This result contrasts with studies with favorable results using another variant of the Stroop: the emotional counting Stroop (ecStroop) [36,37,38]. In this case, participants had to count the number of words with high emotional content which appeared on a screen. It is possible that words, by themselves, are more suitable for persecutory or depressive content than for self-referential content. For the evaluation of the self-referential content, a reference context is needed to differentiate the response latency between groups with and without diagnoses.

Keeping in mind that there are no satisfactory results on the eStroop test with self-referential content words [35], and that there is no test based on the ecStroop paradigm to characterize vulnerability to psychosis, for this study, a variant of the ecStroop (the Testal emotional counting Stroop, TECS), was applied with statements which either place the participant in daily situations related to IRs or are neutral. This design would provide evaluation of individuals vulnerable to psychosis with more ecological validity by presenting statements with referential content (like PLE) that affect the latency of the response due to their strong emotional content.

Accordingly, the objectives of this study were to find out whether individuals vulnerable to psychosis from the general population (Community Assessment of Psychic Experiences (CAPE)-42) [39] would take longer to react (latency) to IR stimuli using the Testal emotional counting Stroop (TECS) (Hypothesis 1). This result would provide support for the interference effect specifically related to IR stimuli among those identified as vulnerable to psychosis, as well as the usefulness of statements with referential content related to everyday situations for the evaluation of vulnerable individuals.

In spite of the enormous importance of emotions in processing information in psychosis [40] and states of vulnerability [17], there have been no previous studies on the role of emotions in the ecStroop paradigm within the scope of vulnerability to psychosis. An implicit test would be useful for observing indicators of vulnerability to psychosis and assessing the presence of emotional and cognitive factors that presumably facilitate the emergence of psychotic indicators (as IR). According to the theoretical models that relate individual vulnerability in interaction with stress, anxiety, depression, and other cognitive variables [41], the second objective of this study is set. Thus, the second objective was to demonstrate whether longer latency in responding to IRs (TECS), observed in individuals from the general population with vulnerability to psychosis (CAPE-42) [39] and high aberrant salience (ASI) [42], would be mediated by the presence of stress [43], anxiety [44], and depression [45] (Hypothesis 2). Furthermore, given the importance of cognitive deficits as indicators of vulnerability in the development of psychosis [46], this hypothesis considers verbal memory as a measure of premorbid IQ (Vocabulary Subtest, WAIS-VS) (in the sense of one of the classic “hold subtests”) [47]. Therefore, premorbid IQ (verbal memory) works as a mediating variable between vulnerability to psychosis and longer latency in response to sentences with referential content.

## 2. Material and Methods

### 2.1. Participants

The sample consisted of 240 subjects aged 17 to 58, with a mean age of 23.97 years (standard deviation (SD) = 4.65), of whom 67.9% were women, at the University of the Republic Litoral Norte Campus (Uruguay). As a condition of participation in another study was not having taken the TECS in order to verify the effects of applying this test, this study analyzed 160 participants who had taken either of the two versions of the TECS (see Section 2.3 Instruments), *M_age_* = 24.12 years (SD = 5.28), of whom 67.5% were women with no history of brain damage, learning problems, severe medical disease, substance abuse/dependence or psychotropic medication. University students who were taking psychology courses were chosen by accessibility. The participants gave their written informed consent before being evaluated. A favorable report was received from the Institutional Research Ethics Committee of the University of the Republic Litoral Norte Campus (Uruguay).

### 2.2. Design

An ex post facto design was used. Subjects were assigned alternately to one TECS version or the other after their evaluation (see Section 2.4 Procedure). The methodology was based on comparison between groups of vulnerable/non-vulnerable participants, comparison of the effect size between the TECS versions and by vulnerability, and the role of emotional functioning in IR response latency.

### 2.3. Instruments

#### 2.3.1. First Self-Reported Evaluation (by Authors)

Collects basic sociodemographic data, history, substance use or medication.

#### 2.3.2. Community Assessment of Psychic Experiences-42 (CAPE-42)

This instrument consists of 42 items which evaluate the positive (20 items), negative (14 items) and depressive (eight items) dimensions of vulnerability to psychosis. The answer format is a four-point Likert scale ranging from never (1) to nearly always (4). When the answers “sometimes” to “nearly always” are chosen, the degree of distress such experiences cause is rated on a four-point Likert scale (from 0 = “not distressed” to 3 = “very distressed”) [39]. The Spanish version shows adequate reliability and validity indicators which support its use as a measure of psychotic phenotype in the general population [48]. In this study, distress was measured using the positive dimension present, which had an internal consistency of 0.96 for the total scale, and 0.83 for the positive dimension.

#### 2.3.3. WAIS–III Test

Spanish version of the WAIS-III test, vocabulary subtest (WAIS-VS). This consists of 33 simple verbal stimuli, words which the subjects must define. Each word scores 0 to 2. It measures the formation of verbal concepts, verbal and semantic richness in the cultural context of the subject. It is applied as a measure of verbal memory or premorbid IQ [49].

#### 2.3.4. Depression, Anxiety and Stress Scales (DASS-21)

This 21-item test evaluates the cognitive, physical and emotional symptoms in the dimensions of depression, stress and anxiety (each subscale with seven items), with four answer choices on a Likert-type scale (from 0 = “did not apply to me at all” to 4 = “applied to me very much or most of the time”) [50]. The Spanish version has adequate psychometric properties, with internal consistency levels varying from 0.73 to 0.81 [51]. For this study, the internal consistency was 0.96 for total items, 0.93 for depression, 0.92 for stress and 0.90 for anxiety.

#### 2.3.5. Aberrant Salience Inventory (ASI)

This is comprised of 29 Likert-type (from 0 = “never” to 5 = “always”) items which measure proneness to psychosis as a characteristic assigning aberrant salience to senses, emotions or thoughts. The test has an internal consistency of 0.89, convergent and discriminant validity in a North American population [42]. For this study, the previous Spanish validation [52] was used. The internal consistency for this study was 0.97.

#### 2.3.6. Testal Emotional Counting Stroop (TECS) (by Authors)

The TECS is an offline desktop application for the storage and management of words and sentence banks which are displayed on a screen based on various time limits and delays. The TECS evaluates response errors, omissions and latency when counting words or sentences of emotional and neutral content. It also records the items that the participant remembers [53]. The TECS was developed in *C++* with the wxWidget toolkit. In this study, the TECS was applied on a x86/x64, 2.33 GHz compatible processor and 1024 × 768 screen resolution. Four versions were developed for the study of vulnerability to psychosis. Version 1 consists of presenting a word that is repeated four, five or six times (the number of words is counted). In version 2, the number of times three-word phrases are repeated (four, five or six repetitions of the same sentence) is counted. In version 3, the number of words in a sentence is increased to four, five or six, and each sentence is repeated at different times during the test (the number of words in each sentence is counted). In version 4, the participant counts the number of words (four, five or six) in sentences that are not repeated during the test. Versions 2 and 4 of the TECS were chosen for this study, and the response latency to the stimulus presented was recorded. Versions 2 and 4 were administered to participants on a device which had only a numeric keypad.

##### TECS Version 2: Different Number of Repeated Sentences

The instructions are to type the number of sentences that appear on the screen as quickly as possible. Participants are given three practice tries. Then four blocks of sentences are repeated at random four, five or six times (Figure 1), with a four-second rest between blocks (during which the phrase “Please wait” appears). The sentences are selected at random from a bank of 110 neutral sentences with 330 words and 131 referential sentences with 393 words. These stimuli were designed based on the main IR evaluation instruments, with varying content (gestures, communication media, coincidences, etc.) adjusted to three-word sentences. The thirty-six neutral and referential content sentences are mixed in the last block of sentences, without pause between them, and always ending with neutral stimuli. When the participants have finished the test, they are asked to write as many sentences as they can remember. These results were not taken into account in this study.

##### TECS Version 4: Unrepeated Sentences with Different Numbers of Words

The instructions are identical to the above except that participants must type the number of words in a sentence that appears on the screen. Four blocks of four, five and six-word sentences are used at random (Figure 2), with a four-second rest between tests (during which the phrase “please wait” appears). The sentences were selected at random from a bank of stimuli with 102 neutral sentences consisting of 511 words and 432 referential sentences consisting of 2157 words. A total of 36 neutral and referential content sentences are mixed in the last block of sentences, without pause between them, and always ending with neutral stimuli. As in the previous version, when the participants finish the test, they are asked to write as many sentences as they can remember.

### 2.4. Procedure

All participants were informed of the purposes of the task, the characteristics of the evaluation and commitment to confidentiality of the results, and gave their written informed consent.

The tests were administered in two parts. In a first meeting, the First Self-Reported Evaluation, CAPE-42, WAIS-VS, DASS-21 and ASI were applied. One week later, participants were given version 2 or 4 of the TECS when they came to be evaluated. This one-week delay between test dates attempted to keep the self-reports from influencing the TECS test, and at the same time, enable the cognitive and emotional state to be related to the TECS test.

Participants who scored the same as or above the 75th percentile on the CAPE-42 positive dimension were considered vulnerable to psychosis when the presence of the item caused distress. Subjects below this percentile were considered not vulnerable.

### 2.5. Data Analysis

Descriptive analyses were performed for age, sex, vulnerability/no vulnerability, and for TECS version applied. The Pearson’s correlations were calculated for vulnerability (positive dimension, CAPE-42), ASI, stress, depression and anxiety (DASS-21), WAIS-VS, and response latency in TECS versions 2 and 4. A factorial ANOVA was done for the CAPE-42 and TECS versions 2 and 4 on IR response latency, calculating the effect size (Cohen’s *d*) to compare the vulnerable/non-vulnerable conditions (Objective 1) and to compare the two versions of the TECS.

For the second objective, a mediation analysis was performed of the independent variables (IVs) (CAPE-42 and ASI) on each dependent variable (DV) (latencies in TECS 2 and 4). Mediating variables (MVs) between the IVs and DV [54], specifically stress, depression, anxiety (DASS-21) and WAIS-VS, were analyzed. Following the Baron and Kenny model [55], the total effect (*C*), the effect of the MVs and the IVs (*a*) and the MVs with the DV (*b*) were estimated. The direct effect *C’* was analyzed to see if it was less than the indirect effect *a* × *b*. The Sobel Test was applied to verify that *C* and *C’* were statistically different using the http://quantpsy.org/sobel/sobel.htm application.

Parallel mediation was analyzed by the Hayes [56] PROCESS version 3.0 macro for SPSS. Model 4 with a 95% CI (Confidence Interval) and 5000 Bootstraps was chosen for each of these analyses. The results were considered significant when “0” was not included in the CI. For the rest of the analyses, the SPSS v20 program was used. All the analyses were accepted with at least *p* < 0.05.

## 3. Results

### 3.1. Preliminary Analyses

Several analyses were performed to check whether the participants in the two TECS versions and those who had not taken either came from comparable populations. There were no statistically significant age differences between versions 2 and 4 of the TECS, or between these versions and those participants who did not take either of them, *F* (2, 237) = 0.46, *p* = 0.632: version 2: *n* = 80, M = 24.36, SD = 6.23; version 4: *n* = 80, M = 23.88, SD = 3.99; neither version: *n* = 80, M = 23.68, SD = 3.05. Neither were there any sex differences between the TECS versions and the condition without TECS, *X_2Pearson_* = 0.497, *p* = 0.780: version 2: *n* = 80 (men: 24, women: 56); version 4: *n* = 80 (men: 28, women: 52); neither version: *n* = 80 (men: 25, women: 55). Similar results were found for CAPE-42 scores among the participants who were not vulnerable, *F* (2, 172) = 1.11, *p* = 0.331: version 2: *n* = 55, *M* = 23.65, *SD* = 2.51; version 4: *n* = 57, M = 23.05, SD = 2.18; neither version: *n* = 63, M = 23.06, SD = 2.63. There were no statistically significant differences in vulnerable participants (CAPE-42) either, *F* (2, 62) = 1.10, *p* = 0.339: version 2: *n* = 25, M = 35.12, SD = 4.30; version 4: *n* = 23, M = 33.65, SD = 2.84; and neither version: *n* =17, M = 34.47, SD = 2.58. Hereafter, the subgroup who did not take the TECS is not considered in this study.

The estimated correlations between IVs (CAPE-42 and ASI), DVs (latency on TECS 2 and 4) and MVs (stress, depression, anxiety, WAIS-VS) were all statistically significant at *p* < 0.01. This statistical condition was also met for the completion of the mediation analysis [55]. Age did not correlate with the dependent variable, so it was not considered as covariance. The descriptive statistics and correlations of variables studied are shown in Table 1.

### 3.2. Vulnerability to Psychosis and Interference in TECS Versions 2 and 4

A factorial ANOVA was done to compare vulnerable/non-vulnerable participants to psychosis and TECS versions 2 and 4 (Table 2). In response to the first objective, the average measurement of interference in referential stimuli was significantly higher among the vulnerable participants, with a lower deviation in their scores and large effect size (*d* = 0.99). When the two versions of the TECS were compared, the overall analysis showed differences indicating longer response latency in version 4, lower deviation in the scores and middle effect size (*d* = 0.45). The interaction between CAPE-42 and TECS version was not significant. The comparison of the vulnerable/non-vulnerable participants in each version, showed a large effect size for TECS version 2 (*d* = 1.03) and version 4 (*d* = 1.08).

### 3.3. TECS Version 2

Figure 3 shows the mediation model with its significant relationships for the second objective. A significant relationship was found for the CAPE-42 (total effect *C* = 0.05, *p* = 0.004), and ASI variables (total effect *C* = 0.01, *p* = 0.006) had a significant relationship with IR response latency. The CAPE-42 IV had a significant relationship with the stress, depression and anxiety MVs (*p* = 0.001), and the ASI IV with stress, depression, anxiety and the WAIS-VS (*p* = 0.001). The variance explained was 71.89%, 67.27%, 60.06%, and 23.55% for stress, depression, anxiety, and WAIS-VS, respectively. Stress and the WAIS-VS had a significant relationship with IRs (*p* = 0.001 and *p* = 0.007, respectively). Anxiety showed a trend toward IR response latency (*p* = 0.055). Finally, the CAPE-42 and ASI variables lost significance when the MVs were included in the model (dashed lines in Figure 3) (direct effect of the CAPE-42: *C’*= −0.01, *p* = 0.705; direct effect of the ASI: *C’*= −0.00, *p* = 0.714). Full mediation occurred through stress in the case of the CAPE-42 and the ASI, and further, the WAIS-VS for the ASI. This result suggests that vulnerability to psychosis is not directly related to IR response latency. More than just an intensification or decrease in the response measured (moderation), stress and lower premorbid IQ must also occur to observe the presence of IR (mediation). The Sobel test was statistically significant (*p* < 0.05) for the relationships of the CAPE-42 with stress (*z* = 4.540), for the ASI with stress (*z* = 4.076), and the WAIS-VS (*z* = 2.370). There was a trend of the CAPE-42 toward the relationships with anxiety (*z* = −1.818, *p* = 0.069). Table 3 shows the *a* × *b* indirect effects for the IVs (CAPE-42 and ASI). This set explained 52.56% of the variance.

### 3.4. TECS Version 4

Figure 4 shows the mediation model. No significant relationship with IR latency was found for either the CAPE-42 or the ASI variables (total effect CAPE-42: *C* = 0.04, *p* = 0.153; ASI: *C* = 0.04, *p* = 0.104). The CAPE-42 variable was related to stress (*p* = 0.001), depression (*p* = 0.001) and anxiety (*p* = 0.001), and the ASI variable was related to stress (*p* = 0.001), depression (*p* = 0.001) and the WAIS-VS (*p* = 0.003). The variance explained was 85.65%, 75.47%, 72.48%, and 22.32% for stress, depression, anxiety, and WAIS-VS, respectively. Stress, depression and anxiety showed a trend toward IR response latency (stress: *p* = 0.050; depression: *p* = 0.083; anxiety: *p* = 0.095). Neither the CAPE-42 nor the ASI were predictors of IR latency when the MVs were included in the model (dashed lines in Figure 4) (direct effect of CAPE-42: *C’* = −0.01, *p* = 0.841; direct effect of the ASI: *C’* = −0.01, *p* = 0.594). No mediation was found in the relationship between the CAPE-42 and ASI IVs with respect to IR latency (although the ASI variable showed a trend). The Sobel test was not statistically significant (*p* > 0.05), but the relationships of the CAPE-42 with stress (*z* = 1.940, *p* = 0.052) and anxiety (*z* = −1.658, *p* = 0.097), and of ASI with stress (*z* = 1.849, *p* = 0.064) and depression (*z* = 1.695, *p* = 0.090) were tendential. Table 4 shows the *a* × *b* indirect effects of the independent variables (CAPE-42 and ASI) on the MVs. This set of variables explained 67.99% of the variance.

## 4. Discussion

Early identification and intervention in psychotic disorders is a priority objective in mental health [57]. Given the usual drawbacks of self-report instruments for evaluation of psychotic-like experiences (PLEs) and ideas of reference (IRs) [26], the implicit Testal emotional counting Stroop (TECS) test, based on the emotional counting Stroop paradigm, was applied. For the first objective of this study, the hypothesis that participants vulnerable to psychosis would show significantly higher response latency to IRs than the non-vulnerable group (large effect size) was tested by applying two versions of the TECS. The result, demonstrating interference, coincides with studies that delusional patients show a bias in their memory of words with threatening content [34,58]. Similarly, an interference effect with words with threatening content has been found using the emotional Stroop (eStroop) paradigm [59]. Similar results have even been found with family members of patients with schizophrenia and bipolar disorder [60]. However, some studies have found inconsistencies, mainly for not differentiating between high and low levels of positive schizotypy [37].

Two versions of the Testal ecStroop (TECS) were applied, one with repeated neutral and referential sentences (version 2, closer to the classic ecStroop: sentences are counted), and another in which sentences were not counted (version 4: words are counted). Differences were found between vulnerable and non-vulnerable participants, with a slight advantage in the effect size for TECS version 4. Perhaps, as its stimuli are not repetitive (possible lower learning effect), and better represent daily contexts, as they are grammatically more complete sentences, it differentiated vulnerable participants well with a lower standard deviation. Participation of the emotional variables and other processes often involved in the study of psychosis (aberrant salience and premorbid IQ) were analyzed for the second objective. TECS version 2 had the most complete results. It was observed that stress mediated the relationship between vulnerability to psychosis and IR response latency. Similarly, the stress and premorbid IQ variables mediated the relationship between aberrant salience and IR response latency. Anxiety showed a trend toward mediation between vulnerability to psychosis and aberrant salience on IR response latency.

Although all the conditions for demonstrating mediation were not met for version 4, the direction of the relationships between variables again point to the mediating role of stress (anxiety and depression to a lesser extent) between vulnerability and response latency to referential stimuli. This version had the highest percentage of explained variance.

These results could be understood in the line of publications suggesting that certain positive clinical conditions, such as suspicion, interact with anxiety and modulate attention to emotionally charged stimuli [61]. Specific cognitive processes in this interaction share erroneous attributions to irrelevant stimuli (aberrant salience), which does not occur in other psychopathological conditions [44].

The mediation analyses suggested in this study, more clearly in TECS version 2, the outstanding role of emotional state, particularly of the stress response, in line with the prediction of appearance of positive symptoms and attenuated psychotic symptoms in vulnerable participants [62].

A novel contribution of this study is that PLEs (such as IRs) may depend on predictor and mediator variables which are not static. Vulnerability to psychosis, as measured by the distress of the CAPE-42 positive dimension (not just its frequency), depends on a state of activation or stress, but not on premorbid IQ, for relating to PLE. A cognitive-emotional state, such as aberrant salience, may repeat a functional schema, but further depend on synergy with premorbid IQ, causing full mediation in both predictor variables. The lower confidence interval for aberrant salience could mean a more precise moment is necessary to characterize the proximity of the PLE.

The role of premorbid IQ in longer response latency, and particularly with aberrant salience, could suggest that the psychotic process has begun when some indicators of cognitive deterioration appear [63,64], especially the relationship between positive symptomatology and working memory [65] or processing speed [66]. The role of the premorbid IQ variable as a mediator in IR response latency has to be related to executive functioning problems in persons vulnerable, at least, to developing psychosis [67].

Although this is a novel study providing a new way for implicit evaluation of IRs, certain limitations should be kept in mind. First, a cross-sectional design was used, so results may not be taken as definitive. The small sample size makes it hard to generalize findings beyond the group selected. Although the differences were statistically significant in favor of subjects vulnerable to psychosis, and there was a certain advantage of TECS version 4, replication is required to consolidate the results. For example, there are some differences between the versions in variables such as aberrant salience, stress or anxiety, which may be influencing the lack of mediation in version 4. The relationships found, the role of vulnerability, and the importance of sensitivity to the referential stimuli proposed in this study must be verified. Also, the possible effect due to multiple comparisons must be taken into account. Furthermore, the study sample was limited to university students, so inferences about other non-clinical samples should be made with caution. However, these findings are a first promising step in the process of predicting the onset of psychosis in a group at psychometric risk. Finally, following studies should include a clinical sample to attempt to elucidate the predictive power of the instrument used in this study, in addition to using different positive and negative symptom criteria.

## 5. Conclusions

This study focused on PLEs as related to subclinical indicators or attenuated psychotic symptoms, a relevant perspective for understanding the development of psychosis [68]. It provides evidence that participants vulnerable to psychosis show longer response latency to referential stimuli (IRs). The two versions of the ecStroop (TECS) used in this study attempt to simulate a context close to reality, in which version 4 may have a certain advantage. Emotional symptomatology, mainly stress, although not anxiety or depression, and a lower premorbid IQ may be risk factors in the process related to vulnerability to psychosis and high aberrant salience [1,43].

## Figures and Tables

**Figure 1 jcm-08-01956-f001:**
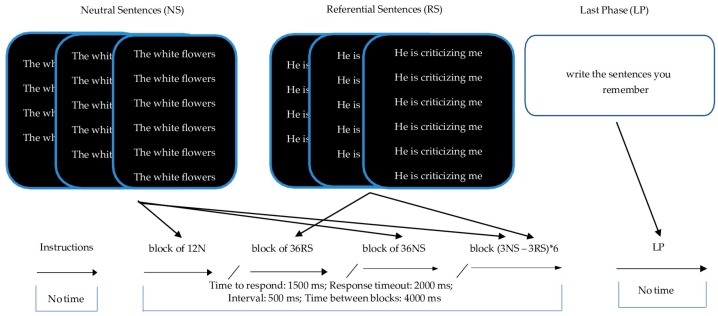
TECS version 2: Examples of how neutral and referential stimuli are presented are shown at the top of the figure. Although translation into English of the examples shown here necessitated four words, referential sentences in the original Spanish version have only three words. In the last phase, the subject is asked to write down the sentences he/she remembers. The center of the figure shows the different blocks of phrases repeated at random four, five or six times. The bottom of the figure shows the response time, time between sentences, between blocks and rest between blocks in milliseconds.

**Figure 2 jcm-08-01956-f002:**
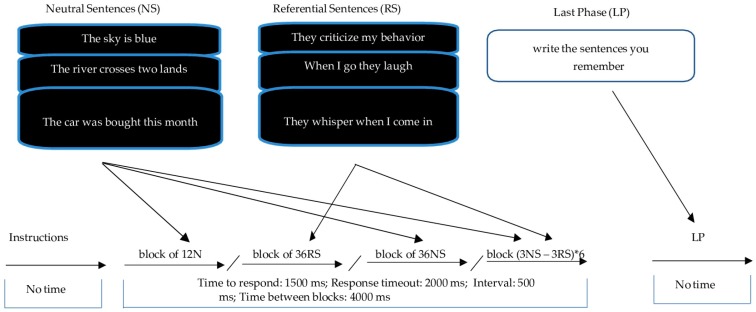
TECS version 4: Examples of how neutral and referential stimuli are presented are shown at the top of the figure. In the last phase, the subject is asked to write down the sentences he/she remembers. The middle of the figure shows the different blocks of four, five and six-word sentences shown at random. The bottom part of the figure shows the response time, time between sentences, between blocks and rest between blocks in milliseconds.

**Figure 3 jcm-08-01956-f003:**
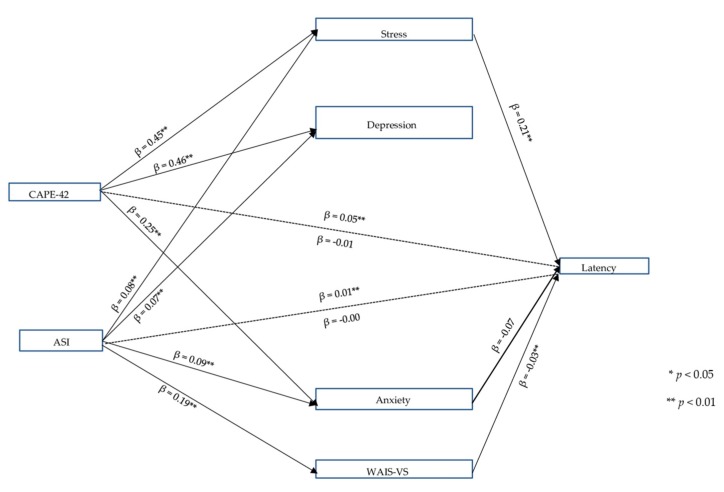
Multiple mediation model of IR response latency. Version 2. Note: β: non-standardized coefficient; CAPE-42: vulnerability to psychosis; ASI: aberrant salience; Latency: Response latency to referential phrases measured with the TECS (*Testal Emotional Counting Stroop*); Stress, Depression and Anxiety: DASS-21; WAIS-VS: Wechsler Adult Intelligence Scale, Vocabulary Subtest. Only total, direct, significant effects and trends are shown. Other results were omitted to facilitate interpretation.

**Figure 4 jcm-08-01956-f004:**
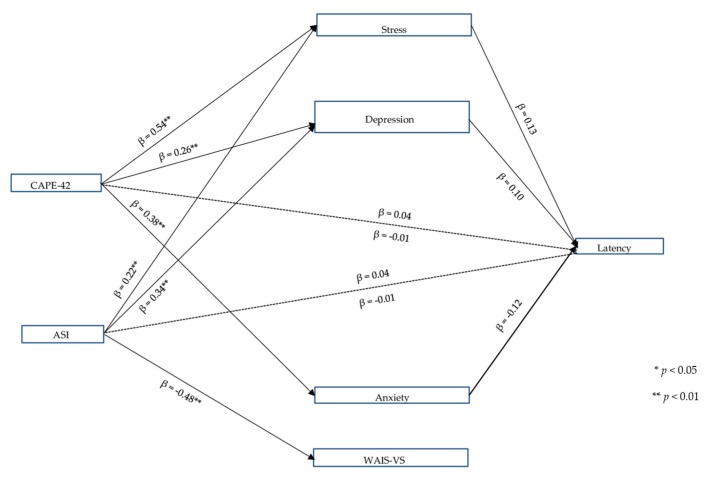
Multiple mediation model of IR response latency. Version 4. Note: β: non-standardized coefficient; CAPE-42: vulnerability to psychosis; ASI: aberrant salience; Latency: Response latency to referential phrases measured with the TECS (*Testal Emotional Counting Stroop*); Stress, Depression and Anxiety: DASS-21; WAIS-VS: Wechsler Adult Intelligence Scale, Vocabulary Subtest. Only total, direct, significant effects and trends are shown. Other results were omitted to facilitate interpretation.

**Table 1 jcm-08-01956-t001:** Descriptive statistics and correlations by participants to whom version 2 was applied (*n* = 80, in bold type), and participants to whom version 4 was applied (*n* = 80).

	1	2	3	4	5	6	7	8
CAPE-42	-	**0.788 ****	**0.899 ****	**0.789 ****	**0.849 ****	**−0.358 ****	**−0.229 ***	**0.446 ****
ASI	0.392 **	-	**0.843 ****	**0.845 ****	**0.706 ****	**−0.472 ****	**−0.123**	**0.453 ****
Stress	0.754 **	0.653 **	-	**0.902 ****	**0.810 ****	**−0.461 ****	**−0.295 ****	**0.553 ****
Depression	0.751 **	.597 **	0.834 **	-	**0.741 ****	**−0.516 ****	**−0.248 ****	**0.554 ****
Anxiety	0.603 **	0.685 **	0.802 **	0.703 **	-	**−0.328 ****	**−0.144**	**0.345 ****
WAIS-VS	−0.185	−0.485 **	−0.324 **	−0.423 **	−0.312 **	-	**0.280 ***	**−0.201**
Age	−0.357 **	−0.154	−0.280 *	−0.269 *	−0.270 *	0.093	-	**−0.182**
Latency	0.436 **	0.423 **	0.652 **	0.514 **	0.422 **	−0.404 **	−0.103	-
M	27.24	21.63	7.13	4.14	3.00	44.43	24.36	1477.14
SD	6.21	23.96	4.75	4.71	3.94	9.39	6.32	157.49
**M**	**26.10**	**8.15**	**5.30**	**3.44**	**1.83**	**46.96**	**23.88**	**1534.75**
**SD**	**5.38**	**7.46**	**4.67**	**4.35**	**2.66**	**7.11**	**3.99**	**90.87**

Notes: 1. CAPE-42: vulnerability to psychosis; 2. ASI: aberrant salience inventory; 3. Stress, 4. Depression, 5. Anxiety: Depression, Anxiety and Stress Scales (DASS)-21; 6. WAIS-VS: Wechsler Adult Intelligence Scale, Vocabulary Subtest; 7. Age of participants; 8. Response latency on referential sentences. * The correlation was significant for *p* < 0.05. ** The correlation was significant for *p* < 0.01. M, mean; SD, standard deviation.

**Table 2 jcm-08-01956-t002:** Factorial analysis of variance CAPE-42 (vulnerability) and TECS version on latency to referential stimuli.

		*n*	M	SD	*F*	*p*	Cohen’s *d*
Vulnerable		48	1585.27	98.05	32.00	0.001	0.99
Non-vulnerable		112	1471.94	129.48			
TECS 2		80	1477.14	157.49	6.25	0.013	0.45
TECS 4		80	1534.75	90.87			
CAPE-42 × TECS					1.47	0.226	
TECS 2	Vulnerable	25	1572.81	107.16			1.03
Non-vulnerable		55	1433.65	158.17			
TECS 4	Vulnerable	23	1598.83	87.42			1.08
Non-vulnerable		57	1508.89	79.24			

Notes: Vulnerable: score in the 75th percentile or higher on the psychotic dimension of the CAPE-42 with distress; Non-vulnerable: below the 75th percentile on the psychotic dimension of the CAPE-42; TECS 2 and 4: Testal emotional counting Stroop, versions 2 and 4. *n*, number of participants; M, mean; SD, standard deviation; *F*, variance; *p*, statistical significance; Cohen’s *d*, effect size.

**Table 3 jcm-08-01956-t003:** Indirect effects of mediator variables on latency in response to referential sentences. Version 2 (TECS).

	β	SE	LLCI	ULCI
Indirect effects through stress				
CAPE-42	0.096	0.022	0.058	0.146
ASI	0.018	0.005	0.009	0.029
Indirect effects through depression				
CAPE-42	−0.017	0.020	−0.062	0.016
ASI	−0.003	0.003	−0.010	0.002
Indirect effects through anxiety				
CAPE-42	−0.018	0.012	−0.049	−0.001
ASI	−0.006	0.004	−0.016	0.000
Indirect effects through the WAIS-VS				
CAPE-42	−0.000	0.004	−0.007	0.007
ASI	0.006	0.002	0.002	0.011

Notes: A Bootstrapping confidence interval (LLCI–ULCI) which does not contain zero shows significant mediation by the mediator and the independent variable, controlled for all mediator and independent variables. DASS-21: Stress, depression and anxiety; WAIS-VS: Wechsler Adult Intelligence Scale, Vocabulary Subtest; CAPE-42: vulnerability to psychosis; ASI: aberrant salience inventory. Level of confidence for confidence intervals: 95%. LLCI: Low Level; ULCI: Up Level. β: non-standardized coefficient. SE: standard error.

**Table 4 jcm-08-01956-t004:** Indirect effects of mediator variables on latency in response to referential sentences. Version 4 (TECS).

	β	SE	LLCI	ULCI
Indirect effects through stress				
CAPE-42	0.071	0.038	−0.011	0.141
ASI	0.029	0.016	−0.004	0.059
Indirect effects through depression				
CAPE-42	0.026	0.031	−0.007	0.111
ASI	0.033	0.023	−0.015	0.077
Indirect effects through anxiety				
CAPE-42	−0.044	0.031	−0.112	0.010
ASI	−0.004	0.010	−0.035	0.008
Indirect effects through WAIS-VS				
CAPE-42	0.001	0.006	−0.014	0.010
ASI	−0.009	0.008	−0.026	0.005

Notes: A Bootstrapping confidence interval (LLCI-ULCI) which does not contain zero shows significant mediation by the mediator and the independent variable, controlled for all mediator and independent variables. DASS-21: Stress, depression and anxiety; WAIS-VS: Wechsler Adult Intelligence Scale, Vocabulary Subtest; CAPE-42: vulnerability to psychosis; ASI: Aberrant salience. Level of confidence for confidence intervals: 95%. LLCI: Low Level; ULCI: Up Level. β: non-standardized coefficient. SE: standard error.

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
