# Peer review of "Vulnerability to Psychosis, Ideas of Reference and Evaluation with an Implicit Test"

_jcm, 2019, doi:10.3390/jcm8111956_

Round 1
Reviewer 1 Report
The authors examine evidence that a task implicitly measuring ideas of references is associated with measures of PLEs. They found evidence that individuals elevated on PLEs showed longer latencies on this task. The study is novel and interesting, however there were a number of issues regarding the methodology and results.
Specific comments for the authors:
Abstract:
-The aims should reflect the overall hypotheses of the study.
Introduction:
-Ideally, the authors would briefly summarize the results using the ecStroop (starting line 79).
-The rationale for hypothesis 2 is not well set up in the introduction. The authors mention that there are “no prior studies on the emotional role in the ecStroop paradigm..” but it is not entirely clear what this means or why this sets up the mediation analyses.
-The authors should justify the use of the verbal memory as a measure of premorbid IQ.
Methods:
-Did the authors include gender or substance use as a covariarate?
-What data cleaning procedures were in places? How were outliers detected and handled? Any multiple comparison correction?
-The statement “As the effect had been evaluated for implementation of the TECS in another study without TECS…” (starting on line 108) is unclear.
-On line 163 it says that the instructions for Version 2 are to: “type in the number of sentences that appear on the screen..” In Figure 1 it was that the instructions are to “write the sentences you remember”. Please clarify (similarly, for Version 4, the instructions in Figure 2 are different than the in-text description).
-Please provide further explanation as to why the questionnaires were administered a week a part from the task?
Results:
-Were the correlations presented in Table 1 similar when analyzed separately by task version?
-Did the authors conduct power analyses to determine what sample size was needed in order to detect an effect using the TECS? N=80 per task version seems a bit small for conducting multiple mediation analyses.
-The statement: “In the interaction between non-significant variables …” (starting line 239) should be clarified.
-The statements “p=.000” should be modified (p<.001).
-For the mediation tests, could the authors provide the proportion mediated for each of the mediation tests? Also, it would ideal to provide bootstrapped Cis.
Discussion:
-The discussion section at times seemed disjointed (e.g., 1 sentence paragraphs), and the authors should more clearly justify their findings in line with previous research and theory.
Reviewer 2 Report
It is a nice paper on the use of the TECS test in detecting vulnerability to psychosis. Aims of the study and hypotheses are clearly presented. Data analyses are adequate. Tables and graphs are useful.
However, I have the following observations:
Introduction:
explain what a discrimination task is provide possible expanations for "no significant differences were found in response latency between patients (mood, personality, and psychotic disorders) and university students [35]"
Materials and methods:
Add some information on the primary study or, at least, some refereces explain why Versions 2 and 4 of the TECS were used and why other versions were exluded Clarify the what are the stressors analyzed by the DASS21
Results:
provide a definition of mediator (and explain why it is different from moderator and modulator)
Additional:
Please, ask a native speaker to revise the English, especially paragraph 2.3.6 which is crucial for the comprehension of the paper but really hard-to-understand
Round 2
Reviewer 1 Report
My only remaining comment is that Hypothesis 2 is still not well set up in the Introduction, specifically the portion about the relationship between vulnerability and IR response latency being “mediated by the presence of stress, anxiety, and depression”. This should be more explicitly set up (eg, in the paragraph beginning l.67).
Author Response
A clarification of the second objective has been included, and a bibliographic reference has been added to support it:
According to the theoretical models that relate individual vulnerability in interaction with stress, anxiety, depression, and other cognitive variables (Bentall, Corcoran, Howard, Blackwood, & Kinderman, 2001), the second objective of this study is set.
Persecutory delusions: a review and theoretical integration
Bentall, R. P., Corcoran, R., Howard, R., Blackwood, N., & Kinderman, P. (2001). Persecutory delusions: A review and theoretical integration. Clinical Psychology Review, 21(8), 1143–1192. http://dx.doi.org/10.1016/S0272-7358(01)00106-4
